# Combining Sea Level Rise Inundation Impacts, Tidal Flooding and Extreme Wind Events along the Abu Dhabi Coastline

Aaron C. H. Chow * and Jiayun Sun

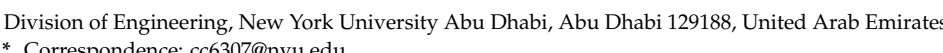

Division of Engineering, New York University Abu Dhabi, Abu Dhabi 129188, United Arab Emirates
* Correspondence: cc6307@nyu.edu

**Abstract:** This paper describes the development of a two-dimensional, basin-scale tidal model with waves and wave run-up to determine the inundation impacts on the Abu Dhabi coastline due to the combined effect of sea level rise, tidal flooding, storm surge and waves. The model combines a hydrodynamics model (DELFT3D), a spectral wave model (SWAN) and wave run-up. A high horizontal resolution (down to about 30 m) is employed in the vicinity of Abu Dhabi—a city built on a system of mangrove islands along the Arabian Gulf coast—to enable prediction of impact at the scale of the local infrastructure, such as individual highway links. The model confirms that, with a rise in sea level of 0.5 m, the islands along the outer coast of Abu Dhabi will experience inundation due to tidal flooding, wind, and high Shamal-induced waves. The incorporation of the wind and waves results in a prediction of more than double the area found underwater within the study area (from 82 to 188 km$^2$). The inner water channel regions of Abu Dhabi, while mostly unaffected by wind-driven wave events, are still vulnerable to tidal flooding. Finally, the paper demonstrates the use of the model to predict whether protection of one segment of the city's coastline will adversely affect the inundation potential of nearby unprotected segments.

**Keywords:** hydrodynamics; Arabian Gulf; coastal inundation; tidal flooding; waves; run-up

## 1. Introduction

There is a global consensus that sea level rise (SLR) has occurred over the 20th century, and will continue through this century, where the mean global sea level could rise about 1 m under the SSP5–8.5 business-as-usual scenario, and as much as 2 m by the year 2100 if Antarctic ice sheet processes are better accounted for [1]. Roughly 360 million people (about 5% of the global population) stand to suffer from the effects of SLR by 2100, with an overall disproportionate effect (75% of the potentially affected population, and nine out of the top ten most at-risk nations) in Asia. Areas under global SLR inundation threat are home to 90% of the current population in nearly 300 urban areas of over 1 million people [2]. Not only does SLR affect inhabitants living on the coastline, but populations living outside the potentially inundated areas are also susceptible to impacts to inland infrastructure, e.g., by reduced accessibility to and from coastal urban areas [3].

As such, the precise identification of areas under SLR threat is a sensible undertaking for urban planning policymakers. The modeling of the combination of tides with SLR has been undertaken in the United States by NOAA, and they have summarized the inundation in a visual form with the Sea Level Rise Viewer [4]. Using this approach for the continental United States [5], it was projected that 13.1 million people would be affected with a 1.8 m SLR by 2100 in coastal communities for the continental United States. In addition, Kulp and Strauss [6] have computed worldwide coastal inundation extents in the CoastalDEM model. Their coastal flooding scenarios were based on the projected mean higher-high water (MHHW) levels in 2100 under the RCP 4.5 climate change scenario. While the visualization tools give significant insight to the areas of the world with inundation risk from SLR, the above analyses use tidal data, using a "bathtub" approach [2,6] whereby

all land areas found below the specified MHHW level, with an added SLR elevation, are defined as under risk. The visualization does not include any potential effects from strong meteorological events that may be superimposed on top of the effects.

A dynamic approach of tidal and sea level rise modeling was performed where wave and wind effects are modeled concurrently with the hydrodynamics in, e.g., in San Francisco Bay for the outer Pacific coast [7]. They showed far larger impacts along the Pacific coast with a combination of waves, tides and sea level rise, compared with sea level rise alone. Furthermore, hydrodynamic modeling of the San Francisco Bay area showed that tidal flooding events can occur far inland, and that different banks of the same water body may experience different effects from the same tidal forcing [8–10].

Increased resolution on the landside has also revealed new areas prone to flooding compared with coarser spatial resolutions (also termed "hyperlocalized SLR impacts" by [5]). For example, by increasing the spatial resolution of the hydrodynamic model along the shoreline of San Francisco Bay, it was revealed that some freeway approaches and some cross-Bay bridge causeways would suffer from inundation, thereby causing a more severe disruption of the transportation network [11,12]. In turn, these impacts on infrastructure could result in some communities of different socioeconomic backgrounds becoming more vulnerable, with implications on social equity [12].

A third and important factor in inundation modeling is the inclusion of storms and extreme weather events along with the SLR and tidal flooding. Morim et al. [13] projected that extreme, persistent wave events will increase in frequency by as much as 100%. Spicer [14] demonstrated impacts from a combination of tidal flooding effects and storms in Maine, but without SLR. Abdelhafez et al. [15] investigated the impact of Hurricane Katrina on the port infrastructure of Mobile, Alabama on the Gulf of Mexico, but also relied on the bathtub approach in their hydrodynamic modeling.

The Abu Dhabi coastline, along the southern coast of the Arabian Gulf, merits additional study in relation to SLR because it is a city comprising a mixture of built urbanized areas with some existing breakwaters, while a large portion of the city is also built on a series of mangrove islands and artificial islands, and, thus, is traversed by numerous tidal channels. Abu Dhabi, like the bulk of the coastline of the United Arab Emirates and the Arabian coast of the Gulf, has a very shallow sloping coastal bathymetry (about 3.5 m per km), meaning that a small rise in sea level can potentially impact areas farther inland [16]. Furthermore, over 85% of the country's population and more than 90% of the UAE's infrastructure is situated within several meters of the present-day sea level [17].

Additionally, the Gulf is known for its very sudden and extreme Shamal wind events— winter northwesterly winds generally occurring in the winter months with winds at about 20 m/s sustained over a period of up to 3–5 days. A typical year will see about 10 Shamal events per year, with an increasing frequency of 1.63 Shamal days per year from 1973 to 2012 [18,19]. Shamal events bring dry desert air to the Gulf, which in addition to causing a tremendous upward evaporative heat loss from the Gulf and severe dust storms, may also raise the local sea level by several meters [19,20]. Inundation impacts due to tides and sea level rise have not been modeled dynamically with local wind and wave effects for the Gulf. Without dynamic modeling with the local tides and winds the predicted inundation extent may either be underestimated (because of the lack of wave effects) or overestimated (because storms may occur during low tide periods).

Numerous previous Gulf models exist at different scales for various purposes: overall circulation patterns [21–23]; long residence time of contaminants in the Gulf [24]; fate and transport of desalination brine discharges into the Gulf [25,26]. To address SLR in Abu Dhabi, Ksiksi et al. [27] used the bathtub approach with a digital elevation model to predict a loss in land area of 60 to 528 km$^2$ resulting from an SLR of 0.5 and 3.0 m, respectively.

AGEDI [28] developed a model to address the potential change of circulation patterns and solute transport within the Gulf, and incorporated Gulf-wide tides and winds with SLR, which does not include the potential for some dry cells to become inundated, i.e., the current coastline was assumed for future scenarios. Vieira et al. [29] have also developed a

wave model to characterize the seasonal wave climate over the entire Gulf, although this does not go on to predict future SLR effects. These models use a coarser spatial resolution than that required for the current analysis. Additionally, the models tend to have a fixed coastline, and do not include wet and dry cells in inland areas that have a potential for flooding from sea level rise, such as that performed by [30] for the San Francisco Bay intertidal areas.

Finally, a motivation of further modeling coastal protection scenarios using a hydrodynamic model will reveal that some protection strategies may result in the worsening of flood impacts to nearby, unprotected communities. This complex interaction of protection has been observed in a large tidal basin such as the San Francisco Bay [3,9].

## 2. Materials and Methods

The coastal model is a combination of three separate models: a hydrodynamic model (DELFT3D) is used to predict tidal water levels and storm surge with SLR; a spectral wave model (SWAN) calculates locally representative wave set-ups; a wave run-up model determines the final run-up water elevation of the highest waves. The schematic of the three near-shore coastal processes are shown in Figure 1—the sum of these three effects is used to predict the overall water level experienced by the local landside infrastructure. The following sections delineate the dynamic coupling of the three models.

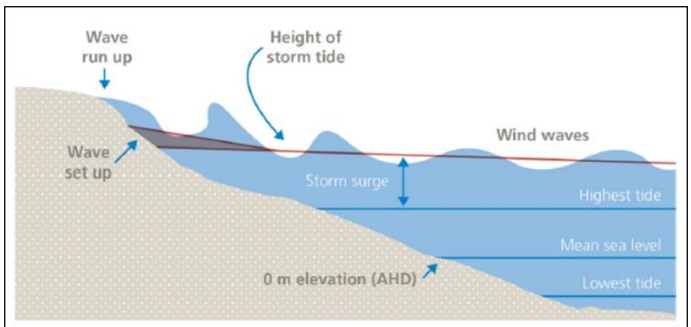

**Figure 1.** Schematic of tidal water level, storm surge, wave set-up and wave run-up (from [31]).

### 2.1. Hydrodynamic Model

The tidal water levels under SLR are predicted using a two-dimensional shallow water Reynolds Averaged Navier Stokes (RANS) model, modified from the Arabian Gulf Community model by Deltares [32,33]. The model domain spans the entire Arabian Gulf (Figure 2), and uses an unstructured grid, which allows horizontal spatial resolutions to range widely from 4 km in the middle of the Gulf to about 30 m along the Abu Dhabi coastline. Additionally, the computational domain stretches several kilometers inland into mainland Abu Dhabi to include cells that could potentially flood (Figure 3).

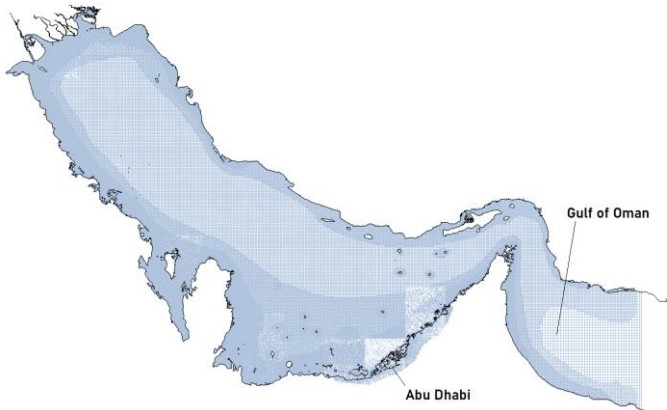

**Figure 2.** DELFT3D AGM Gulf model unstructured grid.

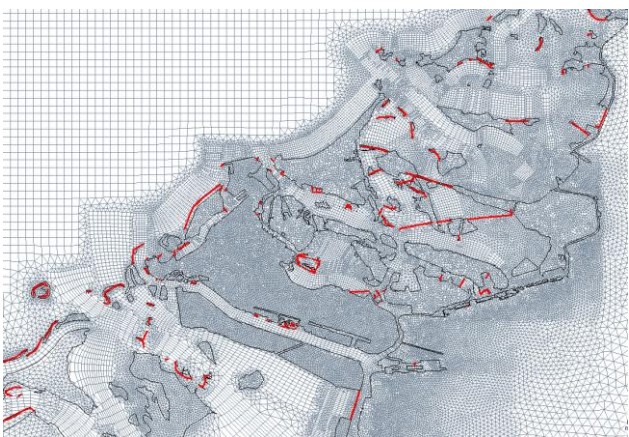

**Figure 3.** Detail of unstructured grid used in current model showing increased horizontal resolution and the use of inland hydrodynamic cells in Abu Dhabi. Black lines show the coastline, and red lines indicate breakwaters already in existence as of 2022.

The model bathymetry was a combination of GEBCO bathymetry data at 15 arc second (450 m) horizontal resolution for most of the Gulf. Near the UAE coastline, the model additionally used a combination of the TanDEM-X digital elevation model (DEM), Landsat-8 and Nautical Charts at 30 m horizontal resolution [34]. A smooth transition between waterside bathymetries and landside elevations is used to factor the possibility of inundation.

Tidal forcings are imposed on the eastern edge of the domain across the Gulf of Oman (Figure 2). The water levels and depth-averaged velocities for the period of interest are taken from the TPXO8 Ocean Atlas, hosted by Oregon State University [35,36]. According to the IPCC AR6 Report, the mean global projected sea level rise by 2100 is between 0.38 m (0.28–0.55 m, likely range) under the SSP1–1.9 scenario and 0.77 m (0.63–1.02 m, likely range) under the SSP5–8.5 scenario. We therefore imposed an SLR of 0.5 m at the open ocean boundary to reflect a projected value that can occur between 2050 and 2100 depending on the climate change scenario used [1].

Atmospheric forcings on the water body are taken from the of ERA5 dataset, an hourly dataset that assimilates global meteorological data from 1979 to the present, that forms part of the Copernicus Climate Change Service Suite [37,38]. As a representation of a strong wind event, the wind field from 1 January–31 March 2017 was chosen as the simulation period because the Shamal (northerly) winds were relatively intense and constant during January 2017. Additionally, Dubai reported flooding in late February 2017 with ~3 m wave heights [39], and the Shamal event caused a meteotsunami event along Iranian coast on 19 March 2017 [40].

### 2.2. Validation of Hydrodynamic Model

The hydrodynamic model is run for a 3-month simulated time period ($\Delta t = 5$ min) without SLR or wind impacts between 1 January and 31 March 2017, and its model outputs for water levels from 10 February to 10 March 2017 at 194 locations throughout the Gulf are used to compare with tidal gauge water level data obtained from the TPXO8 Ocean Atlas for the same period. A measure for agreement between the model and the tidal gauge data is the relative absolute error (RAE), computed for each tidal gauge location by

$$RAE = \frac{\left[\sum_{t=t_0}^{t_1}\left(z_{model}(t) - z_{TPXO8}(t)\right)\right]^{1/2}}{\left[\sum_{t=t_0}^{t_1} z_{TPXO8}(t)\right]^{1/2}} \tag{1}$$

where $z_{model}(t)$ is the model predicted water level at time $t$, and $z_{TPXO8}(t)$ the TPXO8 tidal gauge water level at time $t$. The RAE is calculated at each tidal gauge location using the

time series from $t_0$ = 10 February 2017 to $t_1$ = 10 March 2017. Thus, from Equation (1) a value of RAE = 1 at a location would reflect the performance of a trivial model, and a value of RAE = 0 would signify an exact match between the model and the tidal gauge data for the entire time series. A RAE value closer to 0 reflects a better match between the model and the tidal gauge data.

For the entire Arabian Gulf model domain, the average RAE was 0.71, with 104 of the 194 points (54%) attaining an RAE value of less than 1. Figure 4a depicts the RAE values at locations in the vicinity of the UAE coastline. Figure 4b shows an example of a location on the UAE coastline showing good agreement (Mins Rashid, with RAE of 0.246), and Figure 4c shows an example of a location with fair agreement (Zubbayah Channel, with RAE of 0.484).

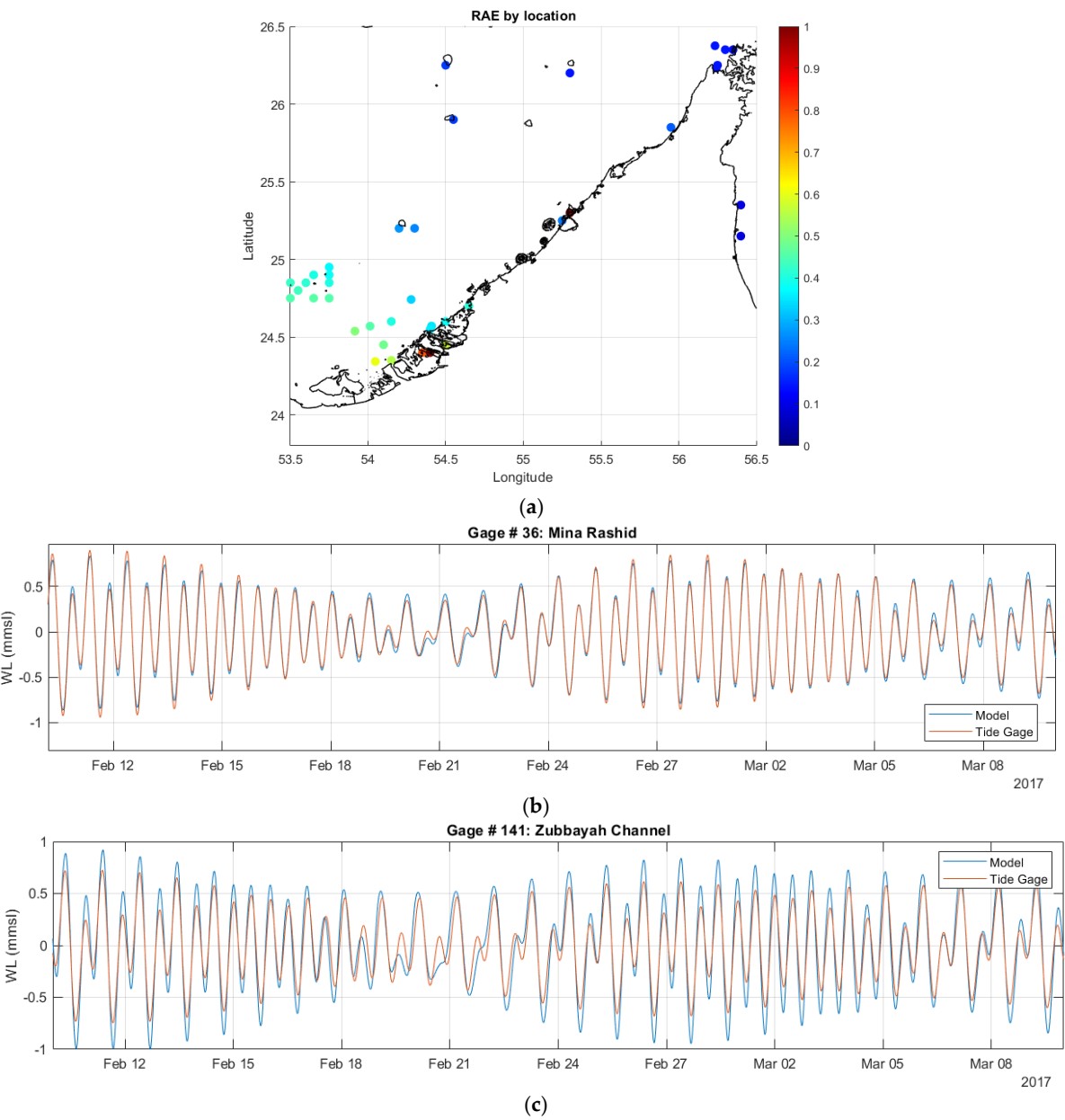

(a)

(b)

(c)

**Figure 4.** Comparison of model output and TPXO8 tidal atlas values: (**a**) RAE computed using Equation (1) and plotted for each tidal gauge location located near the UAE coastline (**b**) Comparison of model output and TPXO8 tidal atlas water level time series from 10 February to 10 March 2017 at Mina Rashid, with RMS error of 0.246 (**c**) Comparison of model output and TPXO8 tidal atlas water level time series for 10 February to 10 March 2017 at Zubbayah Channel, with RMS of 0.484.

### 2.3. SWAN Wave Model

To model the effect of wind generated waves on the Abu Dhabi shoreline, a Spectral Waves Nearshore (SWAN) spectral wave model [41] is used. The SWAN model domain is shown in Figure 5. The SWAN model calculates the significant wave heights, wavelengths and directions of the wind driven waves as they travel across the bathymetry within the domain, considering nearshore processes such as shoaling, wave diffraction, reflection and refraction. The atmospheric forcing of the SWAN model is also obtained from ERA5: air pressure and wind speeds over the domain plus offshore significant wave heights at the domain boundaries.

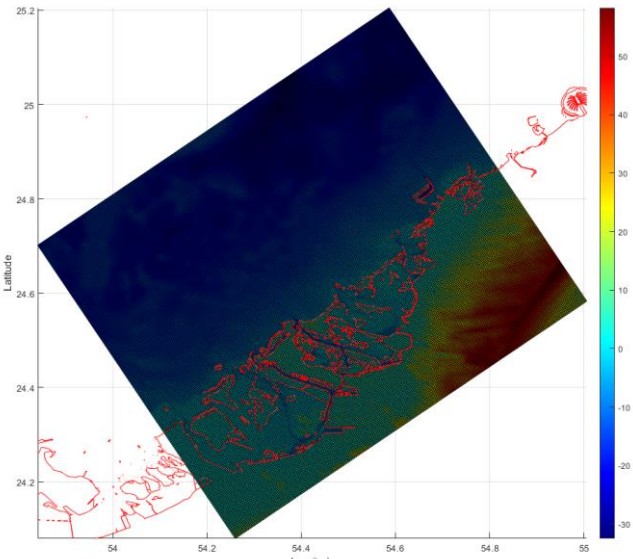

**Figure 5.** SWAN model computational domain, with coastline in red. Shading represents water or land elevation in meters above mean sea level.

The SWAN model was run in nonstationary mode at 5-min time increments over 15 days spanning the strong Shamal event, on a grid aligned roughly parallel and offshore along the outer islands of the Abu Dhabi coastline with a spatial resolution of 0.003 degrees (roughly 300 m). All wave directions and wave frequency ranges of 0.03125–2 Hz are considered.

### 2.4. Run-Up Model

Finally, as data for the nature of the shoreline (i.e., whether the shoreline is natural or constructed) is incomplete or unavailable for the coast of Abu Dhabi, an empirical formula is used to calculate the local wave run-up for natural beaches [42] for the top 2% of run-up events, denoted $R_{0.2\%}$:

$$\frac{R_{0.2\%}}{H_s} = 0.83\xi_f + 0.2 \tag{2}$$

where $H_s$ is the deep-water significant wave height, predicted by the SWAN model, $\xi_f$ is the Irribaren surf similarity number

$$\xi_f = \frac{\tan\beta}{\sqrt{H_s/L_s}} \tag{3}$$

where $\tan\beta$ is the local slope, and $L_s$ the deep-water significant wavelength, also predicted by the SWAN model.

At each time step, the location of the still water level is obtained from the tidal model (by obtaining contour lines of zero water depth from the DELFT3D model).

To determine run-up at each location, local beach slopes were taken from the local bed level bathymetry to the coastline at 300 m intervals along points of the coastline. For each coastline interval, the midpoint of the coastline interval was taken as the point to determine a transect. The significant wave direction computed by the SWAN model at the nearest location to this coastal interval was taken as the orientation of the 1D onshore wave direction. A transect was taken in this wave direction, 150 m on-and offshore of the interval midpoint. Care was taken to ensure this transect does not include any banks from opposite the water body or channel in question. The bed levels along this transect was taken from the bathymetric data also used for the DELFT3D model to calculate the local slope. From the above transects, the significant wave height computed by the SWAN model at the nearest location to the transect was taken as the deep-wave water height for each point where the run up was estimated. Individual wave run-up heights are computed using Equations (2) and (3) at the shoreline at ~300 m intervals along the contour lines. These run-up heights are added to the still water level over a patch of land immediately roughly 150 m alongshore (perpendicular to the SWAN computed wave direction) and 300 m onshore (in the SWAN compute wave direction) of each point.

*2.5. Determination of Inundation Extent*

For each sample point over the hydrodynamic model computational grid, the water level time series was evaluated for connectedness with the tidal and wind signal. Since no overtopping of the levees was assumed, any stagnant water found behind the levees during the computational period were not considered due to inundation but were artifacts from the hydrodynamic model itself. To distinguish points that were inundated from these artifact points, a machine learning technique—k-means clustering—was performed using the method described in [43]. After removing these artifact points that are not connected with the surrounding tidal signal, peak water levels at each point were compared to the nearshore DEM to identify the extent of inundation and peak water depth.

**3. Results**

The combined DELFT3D model was run for a period of 3 months (1 January through 31 March 2017), while the SWAN model was run for the Shamal event (15 February–1 March 2017). Run-up values were computed at each timestep for which SWAN model outputs were available and added to the tidal model water levels by the procedure described in the previous section. At each point of the time series, the peak water level was determined and plotted in Figure 6. Figure 6 shows the inundation extent due to SLR and the tidal model only, compared with the predicted inundation extent incorporating the effects of winds and waves. Whether winds or waves were included, the model shows that inundation occurs on most of the islands of Abu Dhabi, in addition to some areas in the northern and southern mainland portions of the city. Notably, the predicted extent area under water within the study area increased from 82.3 km$^2$ without winds and waves, to 188.4 km$^2$—more than double—with winds and waves. As expected, the increases of inundation with winds and waves were observed on the shoreline areas facing the ocean coastline, but Figure 6 shows that numerous shallow mangrove island areas are also affected by winds and waves.

It is noteworthy that the predicted inundation areas are significantly smaller than that previously predicted for the Abu Dhabi area [6,16]. While it is expected that results from this analysis will differ because of the different climate change scenarios, the current analysis (that includes additional storm effects) yields a smaller extent of inundation on some of the Abu Dhabi mangrove islands as well as the mainland coast to the east of the city. However, their coastal flooding scenarios were based on the projected mean higher-high water level in 2100 under the (RCP 4.5 climate change scenario), without the addition of effects from local wind waves. The current analysis uses the increase of projected water level at the ocean boundary by 0.5 m, within the range of 0.44–0.76 m under the intermediate GHG emissions scenario (SSP2–4.5; [1]).

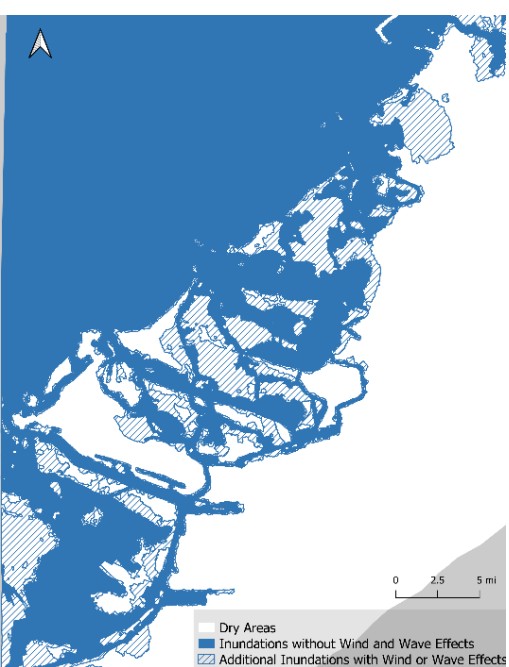

**Figure 6.** Predicted inundation extent for the Abu Dhabi area—SLR with tidal effect only, with and without the effect of winds or waves.

### 3.1. Protection Scenarios

To mitigate the above inundation impact, it is common to install levies and seawalls along the coastline, and Figure 3 shows breakwaters already in place in Abu Dhabi. However, as shown in a large tidal basin such as the San Francisco Bay [3,9], some protection strategies may result in the worsening of flood impacts to nearby, unprotected communities. To investigate whether these complex interactions occur in Abu Dhabi, the current model is run with the same SLR, atmospheric and tidal forcing, but, this time, with added levees to protect each of the 17 individual coastal precincts shown in Figure 7. The 17 precincts are defined roughly based on those proposed in the Plan Abu Dhabi 2030, Urban Structure Framework Plan [44]. The levees are assumed to act as perfect levees with no overtopping. As shown also in Figure 8, a number of protection scenarios consisting of (1) individual precincts (one example shown in Figure 8a), (2) a combination of precincts that are close to each other, such as those on both banks of a tidal channel (Figure 8b) and (3) with all precincts protected (Figure 8c).

### 3.2. Protection Scenario Results

Figure 8 shows the inundation extent due to different individual precinct combinations. Overall, if all the precincts were protected (Figure 8a) some of the enclosed and unprotected mangrove islands may also experience less flooding compared with no protections at all (Figure 8b). Protection of some precincts would sometimes adversely affect the inundation of neighboring precincts. For example, the protection of Precinct 1 (Mussafah) would marginally increase the inundation extent along the neighboring interior coastline of the Abu Dhabi mainland (Figure 8c) while protection of Precinct 13 (Khalifa Port) would serve mainly to protect the Precinct itself without affecting neighboring areas.

Table 1 quantifies these effects of protection of individual precincts on the inundated areas of the other precincts. Each vertical column shows the effect of protecting the precinct on each of the other precincts. For example, in the first column, the protection of Precinct 1 (Mussafah) would increase the inundated area in Precinct 2 by 22% and Precinct 17 (Sas al Nahkl Island) by 17%. These are expected results, because the two affected precincts are located across the water channel from Mussafah. As a result, building a levee to protect Mussafah yields an unwanted effect of more water being diverted into the neighborhoods

located in the opposite banks of the water channel, a phenomenon known as the "levee effect". Finally, Figure 9 graphically depicts the effects of protection of individual precincts on the inundated areas of the other precincts.

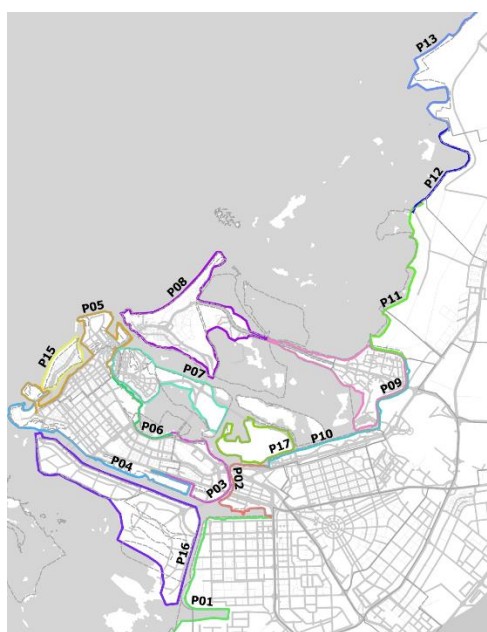

**Figure 7.** Theoretical protection levees for 17 different precincts based on precinct boundaries defined in [44].

**Table 1.** Matrix of the changes in inundated areas in different precincts by protecting individual precinct. The columns represent each protection case. For example, the first column represents changes in inundated areas (expressed as a percentage of the total precinct area) of different precincts by protecting P01—Mussafah. Pink/red shades denote an increase in inundated area for the individual precinct; green shades indicate a decrease.

|  | P01 | P02 | P03 | P04 | P05 | P06 | P07 | P08 | P09 | P10 | P11 | P12 | P13 | P14 | P15 | P16 | P17 |
|---|---|---|---|---|---|---|---|---|---|---|---|---|---|---|---|---|---|
| **P01** | −75% | 1% | 1% | 1% | 2% | 1% | 1% | 0% | 2% | 2% | 0% | 1% | 0% | 0% | 1% | 0% | 1% |
| **P02** | 22% | −86% | 0% | 1% | 2% | 0% | 0% | −1% | 5% | 6% | 0% | 0% | 0% | 0% | 0% | −1% | 0% |
| **P03** | 6% | 0% | −94% | 0% | −3% | 0% | −2% | −3% | 0% | 0% | 0% | 0% | 0% | 0% | 0% | −2% | 0% |
| **P04** | 3% | 0% | 0% | −96% | 0% | 0% | 0% | 0% | 1% | 1% | 0% | 0% | 0% | 0% | 0% | 0% | 0% |
| **P05** | 6% | 1% | 2% | 2% | −91% | 2% | 2% | −1% | 4% | 3% | −1% | 1% | −1% | 0% | 1% | −2% | 1% |
| **P06** | 2% | 0% | −2% | 0% | −1% | −64% | −1% | −1% | 0% | 0% | 0% | 0% | 0% | 0% | 0% | 0% | 0% |
| **P07** | 7% | 1% | 2% | 1% | −3% | 1% | −98% | −3% | 1% | 2% | 0% | 1% | 0% | 0% | 1% | −1% | 1% |
| **P08** | 5% | 1% | 2% | 1% | −1% | 1% | 2% | −98% | 2% | 3% | 0% | 1% | 0% | 0% | 1% | −1% | 1% |
| **P09** | 6% | 1% | 1% | 1% | −1% | 1% | 1% | −2% | −95% | 2% | 0% | 1% | 0% | 0% | 1% | −2% | 1% |
| **P10** | 7% | 0% | 1% | 0% | −2% | 0% | 0% | −1% | 0% | −17% | −1% | 0% | 0% | 0% | 0% | −1% | 0% |
| **P11** | 9% | 3% | 5% | 4% | 5% | 4% | 5% | 5% | 10% | 6% | −95% | 3% | 1% | 1% | 3% | −5% | 2% |
| **P12** | 10% | 1% | 1% | 1% | 6% | 1% | 1% | 1% | 9% | 6% | −1% | −92% | 0% | 0% | 1% | −2% | 1% |
| **P13** | 9% | 1% | 4% | 3% | 6% | 2% | 4% | 1% | 6% | 6% | 1% | 2% | −99% | 1% | 1% | −2% | 1% |
| **P14** | 7% | 0% | 0% | 0% | 6% | 0% | 0% | 0% | 6% | 2% | 0% | 0% | −6% | −98% | 0% | 0% | 0% |
| **P15** | 4% | 0% | 0% | 0% | 0% | 0% | 0% | 0% | 0% | 0% | 0% | 0% | 0% | 0% | −97% | 0% | 0% |
| **P16** | 7% | 1% | 1% | 1% | 2% | 1% | 1% | 0% | 3% | 2% | 0% | 1% | 0% | 0% | 1% | −98% | 1% |
| **P17** | 13% | 2% | 3% | 2% | −8% | 2% | 0% | −7% | 1% | 4% | 0% | 1% | 0% | 0% | 2% | −3% | −96% |

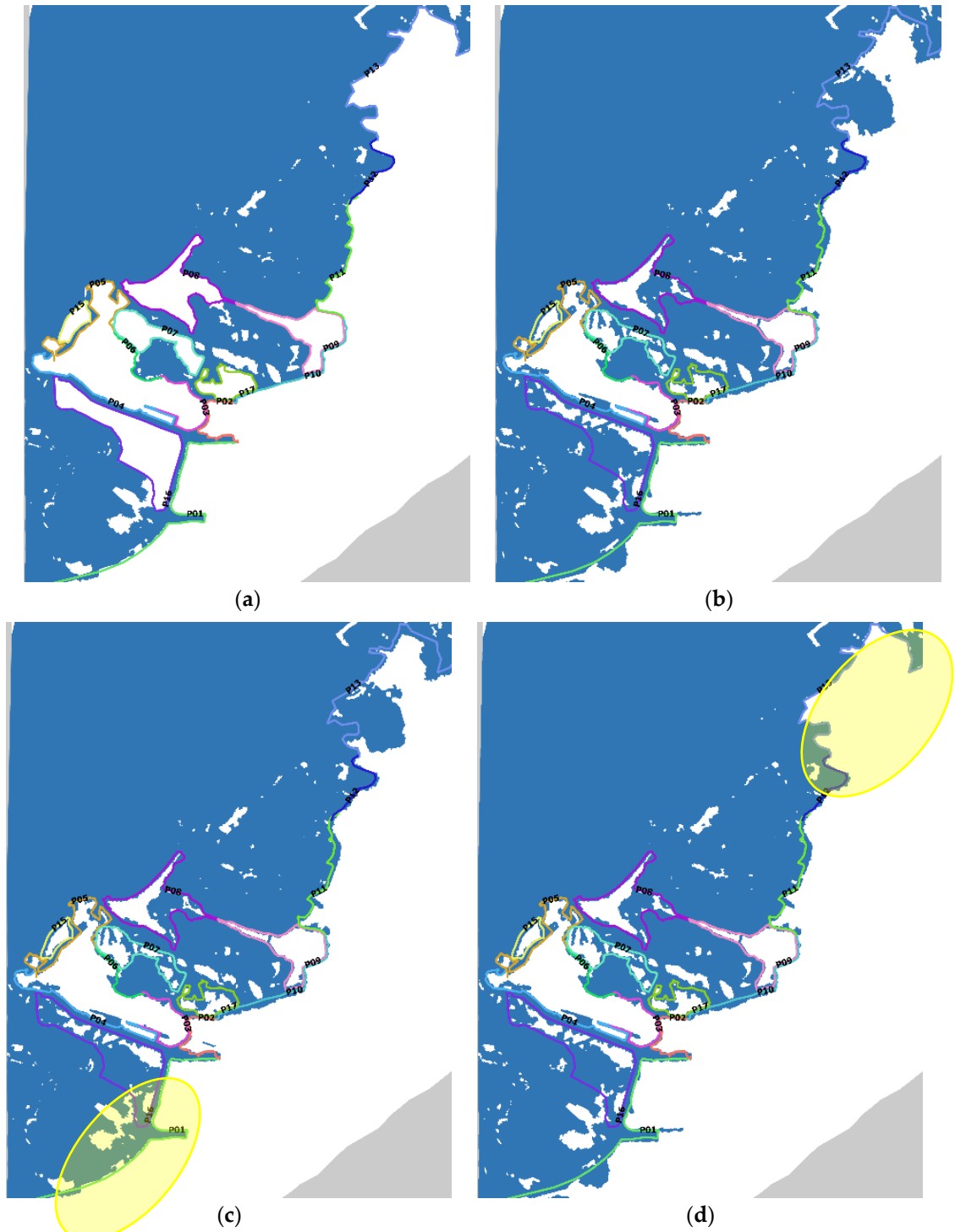

**Figure 8.** Examples of flooding extent predicted by model with various coastal protection scenarios shown as pink outlines: (**a**) all precincts from Figure 7 protected; (**b**) no precincts protected; (**c**) Precinct 1 protected (Mussafah); (**d**) Precinct 13 (Khalifa Port) protected. Colored outlines show the coastal outlines of each precinct. The highlighted areas in (**c**,**d**) represent the protected precincts and the changes of the inundation are observable.

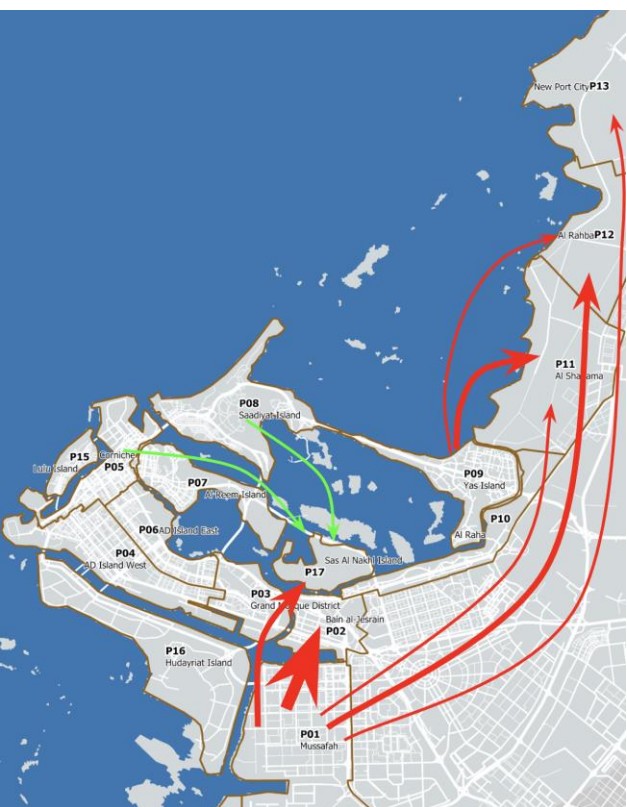

**Figure 9.** Map of the major hydrodynamic interactions from Table 1. Red arrows indicate protecting the origin precinct will increase the inundated area in the pointed precinct. Green arrows indicate protecting the origin precinct will decrease the inundated area in the pointed precinct. The thickness of the arrow represents the magnitude of the impact (by percentage).

## 4. Discussion and Conclusions

A model was developed that combines a Gulf-wide hydrodynamics model (DELFT3D), a spectral wave model (SWAN) and wave run-up model to predict inundation impact on the Abu Dhabi coastline resulting from sea level rise and storm events. The model's high spatial resolution enables prediction of impacts at the scale of the local infrastructure and identification of areas of high inundation risk, either by wind driven waves or tidal flooding. The model confirms that, with a rise in sea level of 0.5 m, the islands along the outer coast of Abu Dhabi, many mangrove islands and many of the northern and southern regions of the Abu Dhabi main coastline will experience inundation. Furthermore, the combination of tidal flooding, wind and high Shamal-induced waves was predicted to more than double the predicted inundation within the study area from 82 km$^2$ to 188 km$^2$. The inner water channel regions of Abu Dhabi, while mostly unaffected by wind-driven wave events, are still vulnerable to tidal flooding events. This predicted inundation extent is smaller than that predicted using a bathtub approach (that identifies all land lower than the MHHW level as inundated), but the inundation extent is somewhat larger in areas vulnerable to storm induced wave run-up.

Finally, the paper demonstrates the use of the model to predict whether protection of one segment of the city's coastline will adversely affect the inundation potential of nearby unprotected segments. While the protection of some precincts such as Khalifa port would not adversely impact neighboring coastlines, protection of certain precincts may have an adverse effect on the inundation of neighboring precincts or those located on the opposite bank of the protected water channel. One notable example is the protection of the Mussafah coastline in Abu Dhabi which may adversely increase the inundation of its neighboring precinct by up to 22%. The levee effect shown in the tidal channels in Abu Dhabi suggests

that both banks of these interior tidal channels would need to be protected for overall protection against inundation.

While this model may identify areas that are of increased vulnerability to either tidal flooding, storm induced flooding or both, it would be necessary to perform more detailed wave run-up and spillover modeling (such as with [45]) while selecting and designing site-specific shoreline protection structures to protect against inundation and storms effectively. Finally, more calibration of the model to further reduce the RAE along the UAE coastline (possibly by adjusting the roughness coefficient of mangrove areas) could be conducted if water level data can be obtained in more nearby locations.

**Author Contributions:** Conceptualization, A.C.H.C.; methodology, A.C.H.C.; software, A.C.H.C. and J.S.; validation, A.C.H.C.; formal analysis, A.C.H.C.; data curation, A.C.H.C. and J.S.; writing—original draft preparation, A.C.H.C.; writing—review and editing, A.C.H.C. and J.S.; visualization, J.S.; All authors have read and agreed to the published version of the manuscript.

**Funding:** This research is conducted under the funding of the NYUAD Transportation Infrastructure Management Lab, led by Samer Madanat.

**Acknowledgments:** This research is conducted under the funding of the NYUAD Transportation Infrastructure Management Lab, led by Samer Madanat. The paper also acknowledges Daniel Sierra for devising coastal protection scenarios, and for the members of the Transportation Infrastructure Management Lab for valuable input on the processing of the model outputs.

**Conflicts of Interest:** The author declares no conflict of interest.

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
