# Peer review of "Combining Sea Level Rise Inundation Impacts, Tidal Flooding and Extreme Wind Events along the Abu Dhabi Coastline"

_hydrology, doi:10.3390/hydrology9080143_

Round 1

Reviewer 1 Report

The manuscript includes the newest sea level rise data and applies the fresh geographic data. The inundation computation has a good accuracy. So, the publication of the material is suitable as a part of Hydrology. But a few revisions are mentioned as follows for better understanding by the readers.

1)     The authors mentioned about the levee protection. Meanwhile, the construction of levee at one point causes the levee effect to another one. What is the final conclusion about the levee implementation?

2)     The authors explain the coastline of Abu Dhabi is consist of Mangrove forest. I think some part is already covered with concrete seawall. Please explain more detail about the present situation of the target coastline,

3)     Figure 3; What is the red line? Authors say it is breakwater. I guess the red lines include the breakwater, seawall and levees.

4)     Figure 4; The figures are not clear.

5)     Figure 6; Scale of (a) and (b) is different? If so, please make the same scale figures. Hopefully, please add the distance meter.

6)     Figure 8; I cannot recognize the difference between the figure (a) to (d).

7)     Line 154; “-3m wave height” What does it mean by (-3m)?

Author Response

Reviewer 1

The manuscript includes the newest sea level rise data and applies the fresh geographic data. The inundation computation has a good accuracy. So, the publication of the material is suitable as a part of Hydrology. But a few revisions are mentioned as follows for better understanding by the readers.

Thank you for taking the time to review the paper.

  • The authors mentioned about the levee protection. Meanwhile, the construction of levee at one point causes the levee effect to another one. What is the final conclusion about the levee implementation?

Levee implementation has been assumed throughout this study. We have elaborated the discussion of how protection of one levee may divert water from one potentially flooded region to a neighboring coastline. It is therefore necessary in cases where both banks of a tidal channel are adversely affected by protecting one of the banks, to protect both sides of the shoreline in question. This finding has been added to the text in the Discussion and Conclusions section of the revised manuscript.

  • The authors explain the coastline of Abu Dhabi is consist of Mangrove forest. I think some part is already covered with concrete seawall. Please explain more detail about the present situation of the target coastline,

Comment noted. It is true that the Abu Dhabi coastline is a mixture of beaches, mangrove islands and developed areas that comprise concrete seawalls and breakwaters. We have elaborated on the section that characterizes the shoreline of Abu Dhabi in lines 72-76 in the revised manuscript.

  • Figure 3; What is the red line? Authors say it is breakwater. I guess the red lines include the breakwater, seawall and levees.

Yes, the reviewer is correct, and we have updated the caption of Figure 3 to reflect that the red lines reflect breakwaters, seawalls and levees that are already existent as of 2022.

  • Figure 4; The figures are not clear.

We have changed our approach for the validation of the model per the suggestion of the second reviewer.  Please see the new version of Figure 4.

  • Figure 6; Scale of (a) and (b) is different? If so, please make the same scale figures. Hopefully, please add the distance meter.

We have made changes to Figures 6(a) and 6(b) accordingly.

  • Figure 8; I cannot recognize the difference between the figure (a) to (d)

We have added some circles to the Figures 8a through 8d to draw the reader’s attention to areas of Abu Dhabi that are either flooded or dry, to make it easier to see the differences of inundation extents in the panels.

  • Line 154; “-3m wave height” What does it mean by (-3m)?

“~3m” denotes “approximately 3 m”. We have updated the text with the latter in the revised manuscript.

Reviewer 2 Report

This paper describes the development of a two-dimensional, basin-scale tidal model with waves and wave run-up to determine the inundation impacts on the Abu Dhabi coastline due to the combined effect of sea level rise, tidal flooding, storm surge and waves. The paper has practical importance for future coastal management in the Gulf Region. In order to be published, however, further revisions are highly required as listed below.

(1) P.4 “Validation”: Not only showing the results in Figure 4 regarding amplitude and phase, more discussions detailed are required with respect to the error in the model so that we can understand the possible error in the numerical model for the future scenarios with sea level rise and protection described in “3. Results” in the present paper. In addition, rather than amplitude and phase, comparison of measured and computed time-variation of tidal level in the same diagram will be more straightforward.

(2) P.5 Run-up model: Basically, Equation (1) has been proposed for a simple straight wave flume with constant beach slope, in which the deep-water wave characteristics can be easily obtained of course. In the present situation, however, the computation has been made for a 2DH (horizontally two dimensional) computational domain. In order to obtain run-up height from Equation (1) at a certain point, how can you obtain the deep-water wave characteristics in the numerical model? How can you obtain the beach slope? More detailed explanation is required how to apply Equation (1) to the complicated configuration in this study area.

(3) P.11 Figure 9: What is the hydrodynamic mechanism of interaction between the protected area and the inundation area? Such an interpretation will be highly useful for similar investigations dealing with protection measures against sea level rise.

Author Response

Reviewer 2

This paper describes the development of a two-dimensional, basin-scale tidal model with waves and wave run-up to determine the inundation impacts on the Abu Dhabi coastline due to the combined effect of sea level rise, tidal flooding, storm surge and waves. The paper has practical importance for future coastal management in the Gulf Region. In order to be published, however, further revisions are highly required as listed below.

Thank you for taking the time to review the paper and for the insightful comments.

  • 4 “Validation”: Not only showing the results in Figure 4 regarding amplitude and phase, more discussions detailed are required with respect to the error in the model so that we can understand the possible error in the numerical model for the future scenarios with sea level rise and protection described in “3. Results” in the present paper. In addition, rather than amplitude and phase, comparison of measured and computed time-variation of tidal level in the same diagram will be more straightforward.

Comment well noted. We have included descriptions and an additional figure to demonstrate the performance of the model compared to a water level time series at tidal gages location along the UAE coastline. We have also included a statistical measure of the performance of the model in the form of the Relative Absolute Error (RAE) in the new Equation 1 of the revised manuscript. Of the 194 tidal gage data available from the TPXO8 atlas, 104 (54%) of the points showed an RAE less than 1, which is considered a better match between the modeled and the measured results compared to a naïve/trivial model, and several comparisons between tidal gage data and modeled results are shown as examples in the new Figures 4a-c of the revised manuscript.

  • 5 Run-up model: Basically, Equation (1) has been proposed for a simple straight wave flume with constant beach slope, in which the deep-water wave characteristics can be easily obtained of course. In the present situation, however, the computation has been made for a 2DH (horizontally two dimensional) computational domain. In order to obtain run-up height from Equation (1) at a certain point, how can you obtain the deep-water wave characteristics in the numerical model? How can you obtain the beach slope? More detailed explanation is required how to apply Equation (1) to the complicated configuration in this study area.

Comment well noted. While we acknowledge that the run-up equation in Figure 1 is applicable for a 1D straight wave flume, we have added some more detail on the method of determining local run-up at various points in the 2D Abu Dhabi coastal domain as follows.

To determine run-up, Local beach slopes were taken from the local bed level bathymetry to the coastline at 300 m intervals along points of the coastline. For each coastline interval, the midpoint of the coastline interval was taken as the point to determine a transect. The significant wave direction computed by the SWAN model at the nearest location to this coastal interval was taken as the orientation of the 1D onshore wave direction. A transect was taken in this wave direction, 150m on-and offshore of the interval midpoint. Care was taken to ensure this transect does not include any banks from opposite the water body or channel in question. The bed levels along this transect was taken from the bathymetric data also used for the DELFT3D model to calculate the local slope. From the above transects, the significant wave height computed by the SWAN model at the nearest location to the transect was taken as the deep-wave water height for each point where the run up was estimated. Equation 1 was then applied using the local slope.

The SWAN model was able to resolve the effects of refraction mentioned by the reviewer. For example, while the prevailing incoming significant wave height in islands exposed to the open sea were on the order of 1 m (and 3 m during the strong storm event), the significant wave heights in the deeper inland channels (say within the Mussafah channel) were only about 0.01 m. Applying Equations 2 and 3 to the transects in these coastal intervals, resulted in very negligible runup predictions (often on the order of 1 mm) in the inland channels, which is consistent with expectations.

We have added in the discussion section that the use of a 2D wave runup model in conjunction to SWAN could be performed as future work on this domain, or as an exercise for another area of interest in the UAE shoreline.

  • 11 Figure 9: What is the hydrodynamic mechanism of interaction between the protected area and the inundation area? Such an interpretation will be highly useful for similar investigations dealing with protection measures against sea level rise.

Thank you for the comment. In short, the protection of one precinct serves to divert water away to neighboring precincts during both high tides and during high storm surge events.

The most notable example of this for the AD coastline is the protection of Mussafah precinct. The protection of the coastline results in water being diverted from Inland Mussafah to precincts immediately downwind and downgradient, such as Precincts 2, 3 and 16 to the north and northeast. We have elaborated the discussion of how protection of one levee may divert water from one potentially flooded region to a neighboring coastline in the Conclusions section (lines 365-371 of the revised manuscript). It is therefore necessary in cases where both banks of a tidal channel are adversely affected by protecting one of the banks, to protect both sides of the shoreline in question.

Round 2

Reviewer 2 Report

None